# Chinese Public Attitudes towards, and Knowledge of, Animal Welfare

**DOI:** 10.3390/ani11030855

**Published:** 2021-03-17

**Authors:** Francesca Carnovale, Xiao Jin, David Arney, Kris Descovich, Wenliang Guo, Binlin Shi, Clive J. C. Phillips

**Affiliations:** 1College of Animal Science, Inner Mongolia Agricultural University, 306 Zhaowuda Road, Inner Mongolia, Hohhot 010018, China; francesca.carnovale@student.emu.ee (F.C.); yaojinxiao@aliyun.com (X.J.); 18686197338@163.com (W.G.); 2Institute of Veterinary Medicine and Animal Sciences, Estonian University of Life Sciences, Kreutzwaldi 1, 51014 Tartu, Estonia; david.arney@emu.ee; 3School of Veterinary Science, University of Queensland, Gatton, QLD 4343, Australia; k.descovich1@uq.edu.au; 4Curtin University Sustainability Policy (CUSP) Institute, Curtin University, Perth, WA 6845, Australia; clive.phillips@curtin.edu.au

**Keywords:** animals, animal welfare, China, attitudes, knowledge, livestock, management, Europe

## Abstract

**Simple Summary:**

Most of our current understanding of attitudes to animals comes from studies conducted in Western countries. China, however, is the world’s biggest producer of farm animals for consumption and has one of the worlds’ largest populations of humans. We conducted a survey of public opinion, in order to better understand Chinese people’s knowledge of animal welfare and their attitudes towards measures to adopt to improve it. Most respondents were unaware of the meaning of animal welfare, but it appears that awareness has increased in recent years. The welfare of wild animals was considered particularly important. The effects of good welfare on the taste and safety of food were highlighted and respondents were willing to pay more for food from animals raised in good welfare conditions.

**Abstract:**

Food-producing animals make up the majority of animals that humans manage globally, and China has been a major producer and exporter of animal products since the late 1990s. The opinions of the population in China regarding animal welfare are not as well understood as those in Europe. In China, animal welfare as a societal concern is still at an early stage of development. This survey of Chinese attitudes aimed to understand consumer knowledge of and behaviour towards animal welfare, and to determine whether harnessing consumer interests may be a potential future influence on the development of high-welfare agricultural production. Most participants were not aware of the meaning of animal welfare, but the number of those that were aware was higher than reported previously. The welfare of wild animals was rated particularly important compared to other animals. The links between welfare and the taste and/or safety of food were considered to be important, and Chinese consumers reported a willingness to pay more for food from animals produced in good welfare conditions, although the quality of the food was considered more important than the animal suffering. A large majority of the respondents reported that there should be legislation protecting animals and certification of welfare on farms, that animals on farms should be provided with enjoyable experiences and that transportation times should be minimised. Furthermore, most respondents reported that animals should be stunned before slaughter. We conclude that animal welfare is of importance to the Chinese consumer, in particular because of its connection to food quality.

## 1. Introduction

In China, as elsewhere, the nuanced differences between animal welfare and animal rights are difficult to understand for the general public [1]. This may be because these concepts were introduced into Mainland China relatively recently, in the early 1990s [1]. Animal welfare can be defined by how well an animal copes with the conditions in which it lives [2]; animal rights are predicated on the idea that the rights of non-human and human animals are, fundamentally, the same [3,4]. However, animal welfare, to a greater extent than animal rights, has attracted increased media attention in recent years [1].

In general, society is becoming more interested in the well-being of animals and our impact on them and the broader environment, at least in Europe [5]. The European Commission for Health and Food Safety [5] reported that 94% of Europeans (including those in the UK) consider it is important to protect the welfare of farmed animals. Within the same report, it was further noted that animal welfare was more important to female respondents than male respondents, and also more important to younger respondents [5]. Food-producing animals make up the majority of animals that are managed by humans globally, and animal farming systems are accused of inefficient use of scarce resources, in particular feed, water and land [6]. Intensive animal production has continued to grow at a rapid rate over the last century [7]. The sustainability of the human-food animal relationship (which includes animal welfare) and the broader environment are likely to be at risk if, as anticipated, prices increase as a result of increasingly scarce feed, water, and land resources on which food animal producers rely [8,9].

China has been a large producer and exporter of animal products since the late 1990s [10,11]. Concerns about China’s record regarding disease control measures and the use of certain proscribed substances in husbandry and food processing have led to a European Union (EU) ban on the import of certain Chinese animal products, with resulting risks to the country’s economy [12,13,14,15]. Chinese livestock industries have experienced a variety of major animal epidemics, such as severe acute respiratory syndrome (SARS), avian influenza, foot and mouth disease and more recently, African swine fever, all of which necessitated large numbers of animals being removed from the supply chain with considerable impact on both the livestock market and animal welfare. Improvement of animal welfare may help to prevent these disease outbreaks [16,17,18]. However, it is suspected that there is a fundamental lack of understanding of the importance of animal welfare among the majority of livestock stakeholders in China, leading to an absence of relevant government policies to address this [19].

Over the past 30 years, China has experienced a growth in affluence, which has been accompanied by a rise in demand for animal products [11] but, in order to improve the welfare of production animals, it is important to understand the attitudes and knowledge of the general public (as consumers) about animal welfare and, in turn, identify potential obstacles to improving the uptake of high welfare products throughout society. Improving animal welfare has direct benefits for the animals themselves, but also has significant benefits for humans who have livelihoods dependent on animal production, and for the wider community in terms of product quality and disease risk management [20].

Currently, little is known about the knowledge and attitudes of the general population towards animal welfare in China. A survey [10] in 2011 revealed that only around one-third of the Chinese public had heard about animal welfare. Of the participants, 73% believed that improving rearing conditions for swine and poultry would improve food safety of meat and eggs, and 54% expressed willingness to pay more for products from welfare-friendly operations. Platto et al. [21] asked Chinese farmers to rate several different priorities for action on farms, for example, provision of better flooring to promote hoof health or better lying areas; the improvement of animal welfare was rated third, with the most important being the farmer’s own well-being [22]. In China, animal welfare, as a societal concern, is still at an early stage of development. It did not attract attention from the Chinese general public until the early years of this century [10]. Many factors are recognized as having an influence on the attitudes of people to animal welfare, including culture, religion and gender [23].

To date, the term “animal welfare” has no meaningful translation in the Chinese language [24,25]. A survey conducted in 2008 found that Chinese respondents had a less favourable attitude towards the importance of typical welfare issues than students in 11 European and other Asian countries [23]; however, in the same survey they had a very favourable attitude towards wildlife protection [23,26]. Student attitudes towards animal welfare are particularly benign in the UK, Sweden and Norway, with females giving higher ratings to animal protection than males [24,26], as well as being somewhat benign in the USA, Japan, France and Germany [27].

This survey aimed to determine the attitudes of the general public in China towards issues that impact on animals, as well as what variables influence their attitudes and their choices. As China is one of the world’s major livestock-producing countries, this survey of Chinese attitudes is important from a global perspective in understanding consumer knowledge and behaviour, and whether harnessing consumer interests can have a potential future influence on the development of high-welfare agricultural production.

## 2. Materials and Methods

### 2.1. Structure of the Questionnaire

The first section of the questionnaire focused on demographic details such as age, gender, level of education, work fields, religious affiliation and place of residence (Box 0). Respondents were then asked how and if they had ever heard of animal welfare and where they had learned about it. Subsequently, they were asked if it was important for them to learn and be taught more about it, or to pay more for animal products with assured good animal welfare, and their opinion regarding the acceptance of good animal welfare by the Chinese population compared to other countries. The rest of the questionnaire was structured in four question sets with answers selected from two 5-point Likert scales. The first question set was concerned with general attitudes towards animal welfare. The second set asked which group of animals they cared most about. The third aimed to determine the reasons that they felt animals should be cared for, the fourth asked what aspects of welfare needed to be most cared for, using the Five Freedoms [28] as the basis for their choices. The survey’s format and content were translated into written Chinese (Zhongwen) by the Chinese authors. The translated version was then back-translated into English for comparison with the original questionnaire and changes were made where discrepancies were evident.

### 2.2. Survey Method

The questionnaire and survey method were approved by the Human Research Ethics Committee of the University of Queensland, Australia (#2019001811). The survey was designed by a cross-cultural research team including researchers from Inner Mongolia, China, and delivered by undergraduate students from the Inner Mongolia Agricultural University (IMAU).

Potential respondents were individually approached in public spaces (e.g., shopping centres, streets, parks, squares, markets) and by door-to-door knocking at residences, as these were likely to be most representative of all members of society. The survey responses (Box 0) were collected anonymously. A total of 217 undergraduate animal science students assisted in questionnaire dissemination and collection, with each distributing approximately ten questionnaires. Thus, a total of 2170 people were approached to complete a questionnaire between August 2019 and September 2019.

Questionnaires were delivered in 23 of the 31 directly administered provinces of the People’s of Republic China, but the majority of responses were from a single province, Inner Mongolia (Figure 1). Questionnaires took approximately 10–15 min to complete. They were delivered in paper form but verbal explanations were also accepted if necessary.

### 2.3. Statistical Analysis

All analyses were conducted using the statistical package Minitab (Minitab Version 18; Minitab Inc., State College, PA, USA). Descriptive statistics were generated and demographic data were analysed to check the differences between responses for all groups (Male; Female; Other; Prefer not to say etc.) using one-way ANOVAs to determine if the answers for different species were significant. Assumptions of normality were checked using the Anderson-Darling test. Non-demographic data were analysed by Ordinal Logistic Regression for ordered categorical dependent variables, and Binary Logistic Regression for binary dependent variables to predict interactions between them.

## 3. Results

A total of 1301 of the 2170 potential respondents completed the questionnaire, a response rate of 60.0%. Demographic responses are shown in Table 1. Respondents were almost equally male and female, while the national average is 3% more males than females, but were skewed towards a younger age (Table 1). About half of the participants were unaffiliated with any religion (atheist 47%), similar to the all-China statistics (51%). High school students (39%) outnumbered the other final levels of education, indicating that survey respondents were educated to a higher level than the all-China levels of education. Approximately 60% of participants were employed full time, and a range of employment fields was represented. The most represented field was agriculture (19%), which in national statistics is only 3% of those employed, and people in military work were least represented (0.8%). Most participants were from urban areas (61%) rather than rural districts or villages, while 58% of the Chinese population live in a rural area. Although most respondents were resident in the province of Inner Mongolia, there were no clear differences that could be attributed to province in our dataset.

### 3.1. Respondents’ Knowledge

The responses to attitudinal questions on animal welfare are shown in Table 2. Almost half of the respondents (47%) had never heard of the term ‘animal welfare’. However, a similar percentage of respondents stated that they live in harmony with animals (43%) and that it is very important to care for animals (53%). About a quarter of respondents stated that animal care should probably not, or definitely not, be taught in schools and only 2% had learned about caring for animals in formal study. Most respondents indicated that they had learned about the care of animals from family and friends or from social media (Table 3).

Most respondents (58%) reported that they would be willing to pay more for animal products if the animals had been well cared for, and more than 60% of these would be willing to pay more than an additional 5% in price (Table 2). More than half of the respondents thought that the current standard of care for animals in China is poor or very poor. A third stated that the standard of animal care in China was similar to other countries, but only 10% responded that it was better or much better. The responsibility for the care of animals was indicated by most respondents to lie with society as a whole (44%), and the number of respondents suggesting it to be mainly the responsibility of farmers was very small (1%).

### 3.2. Attitudes towards Different Animal Taxa

In order to investigate the relative attitudes towards different species, respondents were asked how important it is that different animal groups are cared for (Figure 2 and Table A1). More than 80% thought it was somewhat or very important that mammals, reptiles and birds are well cared for and over 68% responded similarly for insects (Figure 2). In terms of different animal use contexts, the care of pet animals, experimental animals, agricultural animals, stray animals and wildlife were all reported to be somewhat or very important, by over 83% of respondents (Figure 2). Very few respondents answered that being well cared for was ‘not at all important’ for any of the animal groups listed. Respondents considered that it was more important that mammals should be cared for than other animal groups (between *p* < 0.03 and *p* < 0.0001) (Table A2).

Most respondents (>1000) (Table 4) agreed or strongly agreed that reasons to care for animals were for food safety (85%) and for the sake of the environment (85%), and these were more strongly supported than the other options (*p* < 0.05–0.0001): (Table A3 Similarly, most (>900) respondents agreed or strongly agreed that caring for animals makes them feel good (75%), which was more strongly supported than “for the sake of animals” (69%) and “because my religion tells me so” (59%) (between *p* < 0.005 and *p* < 0.0001) (Table 4). Other differences, and their probabilities, are listed in Table A3.

### 3.3. Attitudes towards Animal Welfare and Procedures Performed on Animals

Importance ratings for the evaluated welfare assessment criteria are shown in Table 5. For each criterion the majority of respondents (over 80% in all cases) reported that they strongly supported it, with physical fitness being the most important. Differences between respondents’ answers both within and between criteria are listed in Table A4.

Responses regarding animal procedures are listed in Table 6. A large majority of respondents agreed or strongly agreed that there should be legislation protecting animals, that farms should be certified by animal protection organisations, and that such organisations are important in ensuring these animals’ care. Over half of the respondents considered management mutilations, such as castration, ear tagging and tail docking, to be acceptable. Minimisation of animal transportation time was thought to be important by 79% of respondents. A similar number agreed or strongly agreed that animals should be provided with enjoyable experiences on farms (82%). However, 70% of respondents agreed that it is acceptable for animals to suffer if the quality of the product is good enough, and over a third (44%) if the price of the product is low enough. However, a large majority of respondents thought that animals should be stunned before slaughter and that animals should be dead before being cooked.

## 4. Discussion

The survey we conducted suggests that there has been an improvement in the perception of animal welfare in China since a 2008 survey of students found that China had the lowest acceptance rating for animal welfare issues of 13 Eurasian countries [23,26]. However, that survey also found that there was considerable support for wildlife protection within China [26].

### 4.1. Respondents’ Knowledge about Animal Welfare

Almost half of the respondents had never heard of the term “animal welfare,” which does not necessarily mean that Chinese people do not care about the well-being of animals but Phillips et al. (2012) [26] showed that respondents in a sample of European countries generally had greater concern for the welfare of animals than those in a sample of Asian countries, including China. The Chinese government considers it necessary to adopt intensive rearing in order to meet the growing demand for the products of livestock [24,32,33,34]. As has also been shown in other studies, respondents were very sensitive about killing animals and all practices used on the farm [10]. Respondents mostly knew about animal care and welfare from family and friends, and also from the media. This indicates that reporting in the media may have improved since You et al. (2014) [10] claimed that discussion of animal welfare by the Chinese media was poor at that time. Respondents in the current study mostly felt that they lived in harmony with animals, which may be a reflection of the provinces where the survey was conducted, where agriculture in the economy and animal production are important. Current profession may be more pivotal than educational background in approaches to welfare measures and criteria [9].

Most respondents agreed that it was either very or extremely important to care for animals. Among other reasons, food safety was a common reason for this, as has been found in other studies [5]. Three-quarters of the respondents said that animal welfare should be taught in schools, and likewise Europeans (87%) consider that this is a good way to influence the attitudes of the younger generation towards animals [35,36]. As the survey was distributed by students it is possible that a disproportionate number of the respondents were from high school and university, and educational background influenced views on animal welfare aspects, as has also been shown in other studies [10]. The findings may therefore be skewed towards the perceptions of the younger generation.

The respondents thought that the current standard of care for animals in China is poor or very poor, acknowledging perhaps that there is difficulty in applying high welfare animal husbandry for the production of a large amount of animal products [37]. According to research carried out on meat consumption in China, future spending on meat is expected to increase [38]. This nutritional transition is a response to changes in lifestyle and dietary patterns driven by urbanization, globalization and economic growth, and their resulting impacts on nutrition and health outcomes [39]. But there remains significant diversity of diets around the world, reflecting diversity in food production landscapes and ecosystems, socio-economic conditions, cultures and beliefs. Studies of food systems adapted to their local context, and of the associated traditional knowledge built up over millennia, can provide new insights and pathways towards more sustainable food systems [40]. Most respondents said that they would be willing to pay more for high welfare standard products, which was not found in a previous survey in China [10]. If true, this could drive improvements in good practices on livestock farms; 58% of UK customers believe that by paying more for higher welfare products they can influence the welfare conditions of the animals [41]. In another European survey, Bozzo et al. [42] showed that 58.4% of the persons interviewed would pay 20% more than normal for high welfare products, while in this study 35% of respondents were prepared to pay more than 10% extra, which was most likely due to the perceived improved taste of the animal-derived product and effects on the environment. The European Commission for Health and Food Safety [5] reported that a sample population from 15 Member States of the EU considered that animal welfare contributes to a better-quality animal product.

### 4.2. Chinese Attitudes towards Animal Taxa and Reasons for Care of the Animals

The Chinese population appears concerned about all types of animals, since none of the species listed in the questionnaire were identified by many as unimportant. Davey and Wu [43], reported that Chinese students were concerned about the use of animals for research, which was also found in our study. Interestingly in the current study, wild animals had the highest amount of support from participants: 46% for very important and 39% for somewhat important. This importance attached to wildlife confirms an earlier study in which Chinese respondents did not care much about animal welfare generally [26] but were very concerned about wildlife protection [23,44]. This was further borne out by the findings of Phillips et al. (2012) [26] that of a range of countries, Chinese respondents scored lowest for animal welfare generally, but highest for the importance of welfare issues among wild animals. That this strength of comparative interest in the welfare of wild animals may have a cultural basis is worthy of further consideration and investigation. It may also be due to an increase in information regarding diseases that can be transmitted from wild animals, which up until recently few people were aware of [45]. Consumers consider farm animal welfare as an attribute of the food quality concept, with more importance given to this than to other attributes [46,47]. There is evidence from this survey that the Chinese population has responded positively to understanding the reasons why animals should care for, and how animal welfare affects other aspects, such as food safety, in China. The disease burden and use of antibiotics in farm animals is taken very seriously in China by government and could be considered a platform from which to advocate improvements to animal welfare [48].

### 4.3. Chinese Attitudes towards Animal Welfare and Procedures Performed on Animals

China has not yet enacted animal welfare legislation and the reason for this may be in part due to the perceived lack of animal welfare information in the country [1]. In 2005, the National People’s Congress voted on the Animal Husbandry Law of the People’s Republic of China, but the omission of the term ‘animal welfare’ reflects the fact that much of the public and many legislators are of the opinion that animal welfare cannot become a topic codified in the law [49]. The culture in a country can affect perceptions of animal sentience, which according to several studies [5,26,49] will then correlate with the perception of whether practices involving the animal species are considered cruel or not.

The majority of participants in our study considered the absence of injury to be somewhat important. In the EU, inflicting pain and injury are thought to be so well-controlled that people assume that they must be necessary otherwise they would not be allowed [26]. In this case the European respondents may be more trusting of animal production practices and animal welfare than their Chinese peers.

The respondents generally agreed that animals should be dead before being eaten, and this is evidence to encourage efforts to outlaw the consumption of live animals to reduce suffering and improve animal welfare [26].

The Eurobarometer survey (EC 2007) [5] of the European Commission for Health and Food Safety found that 60% of European respondents believed that welfare protection had improved in their country. In China, the attitude part of the survey appears to suggest that the general public mostly support the promotion of animal welfare.

## 5. Conclusions

The majority of the respondents to our survey remained unaware of the meaning of the term ‘animal welfare’ but the numbers of those that were aware appear to have increased compared with previous studies. Although those that were aware expressed opinions that were positive towards the welfare of animals, the majority considered the care of animals in China to be poor. The role of the popular media in discussing the welfare of animals seems to have improved recently. The respondents that were concerned for the welfare of animals were concerned for the welfare of all taxa and all types of commercial animal uses. A particularly interesting finding, and one that confirms a previous study, was the higher value placed on the welfare of wild animals than for other types of animal uses. The survey also showed the importance given to the taste of food and the safety of food from farm animals, and any possible link these might have to the welfare of the animals used; respondents reported that they would be prepared to pay more for such food.

## 6. Limitations

The authors recognize that there were limitations of this study that may restrict the conclusions that can be drawn. The respondents were not necessarily typical of the population of China as a whole, being more evenly matched to the student administrators of the survey, in terms of gender, age and having a higher education level. Likewise, the respondents were more urbanised in this study than the population of China as a whole. This may have been due to the use of student questioners rather than professional market research questioners, and also the sites selected to carry out the questioning. Finally, narratives related to the welfare of animals that might have been important but not predicted by the designers of the questionnaire may have been missed.

## Figures and Tables

**Figure 1 animals-11-00855-f001:**
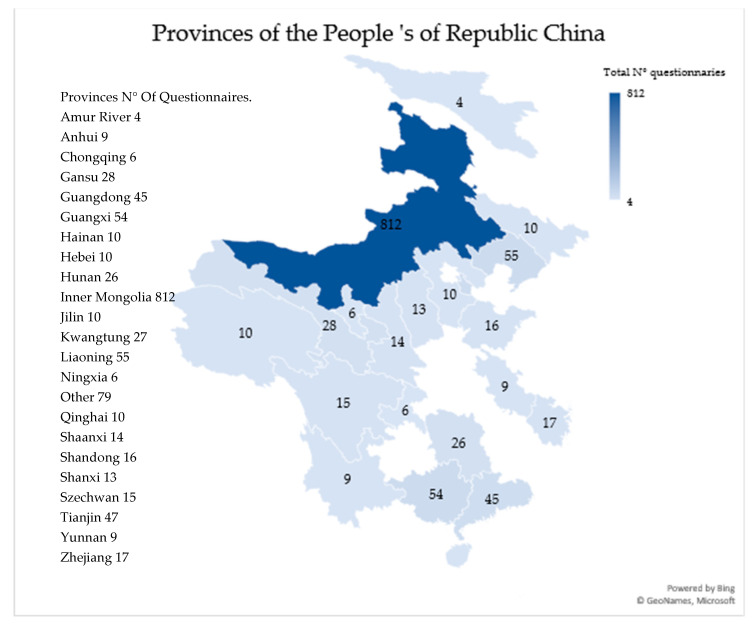
Map showing the collection points from the 23 provinces of the People ‘s of Republic China and the number of questionnaires collected (total N of questionnaries = 1301) [29].

**Figure 2 animals-11-00855-f002:**
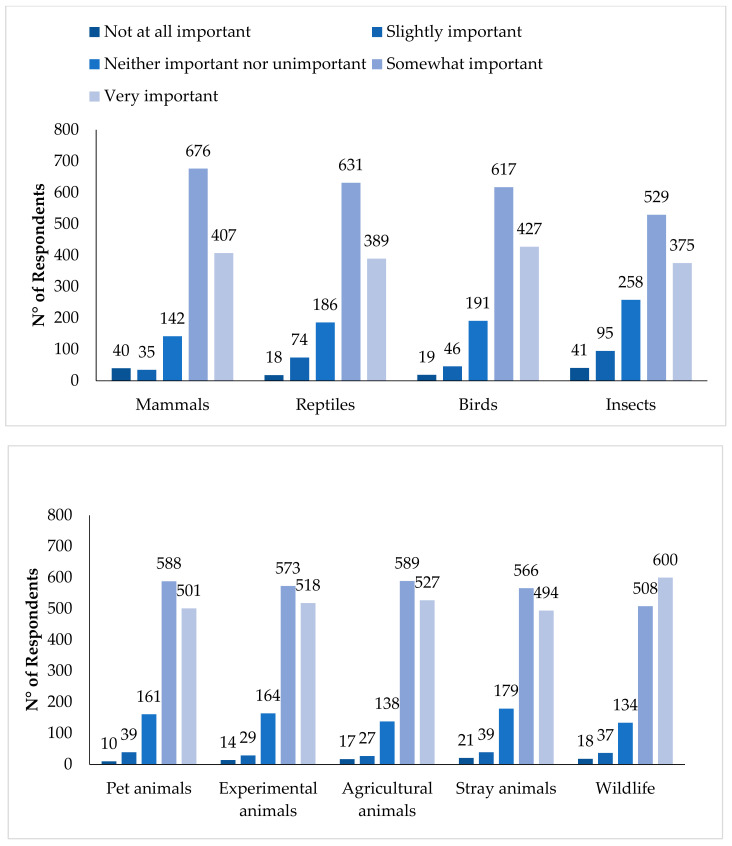
The relative perceptions of attitudes towards animal taxa and different animal-use groups in China, on a scale from “Not important” to “Very important” that they are well cared for.

**Table 1 animals-11-00855-t001:** Demographics of respondents found in the questionnaires analysed. China statistics data from the 2018 China statistical yearbook [30,31].

Demographic Variables		Number of Respondents	% of Survey Sample	China Statistics 2017, n × 10^6^
Gender	Male	621	47	Male 711 (51.17%)
	Female	631	48	Female 678 (48.83%)
	Other	7	0.5	*
	Prefer not to say	39	3	*
Age	18–24	434	33	132 (14%)
	25–34	339	26	189 (20%)
	35–44	253	19	170 (18%)
	45–54	177	13	202 (21%)
	55–64	65	5	127 (13%)
	>65	27	2	130 (14%)
Religion	Chinese folk	213	16	304 (21%)
	Atheist	611	47	720 (51%)
	Buddhism	132	10	254 (18%)
	Muslim	29	2	28 (2%)
	Christians	23	1	72 (5%)
	Daoism	26	2	*
	Confucianism	25	1	*
	Prefer not to say	128	9	*
	Other	106	720	9 (<1%)
Education	Elementary school or below	131	10	730 (55%)
	Technical college	146	11	*
	Middle school	160	12	286 (21%)
	High school	507	39	321 (24%)
	University undergraduate	270	20	0.8 (<1%)
	University postgraduate	86	6	0.5 (<1%)
Employed	Yes	781	60	776
	No	504	39	
Work field	Administration	113	9	*
	Agriculture	239	19	2.25 (3%)
	Arts	44	3	*
	Construction	94	7	26 (33%)
	Education	121	9	17 (22%)
	Finance	39	3	6.88 (9%)
	Government	54	4	*
	Health	78	6	8.97 (11%)
	Mining	22	1	4.55 (6%)
	Military	11	0.8	*
	Retail/Sales	101	8	8.42 (11%)
	Science	23	1	4.20 (5%)
	Technology	65	5	*
	Other	251	20	*
Dwelling	Rural	171	13	813 (58.5%)
	Village	321	24	*
	Urban	793	61	576 (41.45%)
	Other	14	1	*

* = no data.

**Table 2 animals-11-00855-t002:** Respondents’ attitudes towards animal welfare in China.

Questions and Response Options		Number of Respondents	% of Survey Sample
Have you heard of the phrase “animal welfare”?	Not sure	99	7
	Never	608	47
	A few times	453	35
	Many times	128	9
Do you live in harmony with animals?	Not at all	70	5
	Slightly	247	19
	Moderately	411	31
	Very much	312	24
	To a great extent	256	19
How important is caring for animals to you as a person?	Not at all	47	3
	Slightly	176	13
	Moderately	380	29
	Very	471	36
	Extremely	221	17
Do you think that animal care should be taught in schools?	Definitely not	90	6
	Probably not	238	18
	Possibly	477	36
	Probably	308	23
	Definitely	185	14
Would you be willing to pay more for products from animals that are better cared for?	Yes	757	58
	No	532	41
If yes, how much more would you be willing to pay for a product from an animal very well cared for compared with the standard product?	5%	423	35
	10%	328	27
	20%	262	21
	50%	115	9
	100%	36	2
	>100%	41	3
What do you think is the current standard of animal care in China?	Very poor	128	10
	Poor	557	43
	Satisfactory	383	30
	Good	164	12
	Very good	40	3
How do you think the standard of animal care in China compares to other countries?	Much worse	263	20
	Somewhat worse	473	36
	About the same	428	33
	Better	91	7
	Much Better	42	3
Who do you think is most responsible for the adequate care of animals?	Government	100	8
	Animal Protection Organizations	157	13
	Farmers	18	1
	All of society	516	44
	People who like animals	123	10
	People who own animals	167	14
	Companies that use animals	23	2
	Other	48	4

**Table 3 animals-11-00855-t003:** Origin of respondent’s awareness of caring for animals.

Did the Following Help You to Learn about Caring for Animals?		Number of Respondents	% of Survey Sample
Formal study	Yes	29	2
	No	1273	97
Family and friends	Yes	459	35
	No	845	64
Media	Yes	252	19
	No	1050	80
Business	Yes	57	4
	No	1245	95
My job	Yes	110	19
	No	1192	80
Government	Yes	46	3
	No	1256	96
Animal protection organization	Yes	178	13
	No	1124	86
Social media	Yes	359	27
	No	943	72
Farmer	Yes	84	6
	No	1218	93
Have not learnt	Yes	106	8
	No	1196	91
Other	Yes	22	1
	No	1280	98

**Table 4 animals-11-00855-t004:** Reasons for caring for animals, listed in declining order of agreement.

Indicate How Strongly You Agree or Disagree with the Following Reasons		Number of Respondents	% of Survey Sample
It is important for food safety	Strongly disagree	42	3
	Disagree	56	4
	Neither agree nor disagree	97	7
	Agree	673	51
	Strongly agree	433	33
It is important for the environment	Strongly disagree	13	1
	Disagree	51	3
	Neither agree nor disagree	131	10
	Agree	628	48
	Strongly agree	477	36
To improve product quality or taste	Strongly disagree	20	1
	Disagree	34	2
	Neither agree nor disagree	156	12
	Agree	599	46
	Strongly agree	491	37
It is good for human health	Strongly disagree	19	1
	Disagree	61	4
	Neither agree nor disagree	209	16
	Agree	593	45
	Strongly agree	419	32
To improve profit from animals	Strongly disagree	55	4
	Disagree	76	5
	Neither agree nor disagree	178	13
	Agree	576	44
	Strongly agree	416	31
It makes me feel good	Strongly disagree	14	1
	Disagree	63	4
	Neither agree nor disagree	237	18
	Agree	600	46
	Strongly agree	387	29
For the sake of the animals	Strongly disagree	50	3
	Disagree	117	8
	Neither agree nor disagree	225	17
	Agree	514	39
	Strongly agree	395	30
My religion tells me to	Strongly disagree	51	3
	Disagree	113	8
	Neither agree nor disagree	361	27
	Agree	463	35
	Strongly agree	313	24

**Table 5 animals-11-00855-t005:** Attitudes towards animal care based on animal welfare evaluation criteria, in declining order of importance.

How Important are the Following Conditions in Animal Care?		Number of Respondents	% of Survey Sample
Physical fitness	Not at all important	6	0.4
	Slightly important	25	1
	Neither important nor unimportant	95	7
	Somewhat important	576	44
	Very important	598	46
Absence of disease or injury	Not at all important	5	0.3
	Slightly important	23	1
	Neither important nor unimportant	97	7
	Somewhat important	611	46
	Very important	565	43
A comfortable environment	Not at all important	10	0.7
	Slightly important	24	1
	Neither important nor unimportant	131	10
	Somewhat important	613	47
	Very important	521	40
Species-relevant nutrition	Not at all important	37	2
	Slightly important	32	2
	Neither important nor unimportant	96	7
	Somewhat important	661	50
	Very important	475	36
Access to drinking water	Not at all important	8	0.6
	Slightly important	50	3
	Neither important nor unimportant	116	8
	Somewhat important	638	49
	Very important	487	37
Space	Not at all important	4	0.3
	Slightly important	39	3
	Neither important nor unimportant	116	8
	Somewhat important	596	45
	Very important	545	41
Absence of fear or distress	Not at all important	14	1
	Slightly important	40	3
	Neither important nor unimportant	124	9
	Somewhat important	596	45
	Very important	527	40
Absence of pain	Not at all important	10	0.7
	Slightly important	42	3
	Neither important nor unimportant	129	9
	Somewhat important	544	41
	Very important	575	44
Control over their environment	Not at all important	15	1
	Slightly important	40	3
	Neither important nor unimportant	149	11
	Somewhat important	555	42
	Very important	542	41
Opportunity to perform natural behaviours	Not at all important	8	0.6
	Slightly important	42	3
	Neither important nor unimportant	181	13
	Somewhat important	564	43
	Very important	505	38

**Table 6 animals-11-00855-t006:** Attitudes towards strategies for the management of animals.

Indicate Your Level of Agreement with the Following Statements		Number of Respondents	% of Survey Sample
Farms with animals should be certified by animal protection organizations	Strongly disagree	62	4
	Disagree	39	3
	Neither agree nor disagree	133	10
	Agree	660	50
	Strongly agree	406	31
Procedures performed on animals such as ear tags, castrations and tail docking are acceptable for management	Strongly disagree	109	8
	Disagree	245	18
	Neither agree nor disagree	180	13
	Agree	521	40
	Strongly agree	246	18
Transportation time of live animals should be minimized	Strongly disagree	16	1
	Disagree	31	2
	Neither agree nor disagree	211	16
	Agree	641	49
	Strongly agree	400	30
Animals on farms should be provided with enjoyable experiences	Strongly disagree	19	1
	Disagree	31	2
	Neither agree nor disagree	175	13
	Agree	642	49
	Strongly agree	434	33
It is OK to buy products of animals that have suffered if the product quality is good enough	Strongly disagree	178	2
	Disagree	250	6
	Neither agree nor disagree	223	19
	Agree	409	44
	Strongly agree	241	26
It is OK to buy products of animals that have suffered if the price is low enough	Strongly disagree	187	14
	Disagree	277	21
	Neither agree nor disagree	244	18
	Agree	247	26
	Strongly agree	245	18
Animals should be unconscious (stunned) before they are killed	Strongly disagree	34	2
	Disagree	89	6
	Neither agree nor disagree	250	19
	Agree	582	44
	Strongly agree	346	26
Animals should be killed before being cooked	Strongly disagree	30	2
	Disagree	48	3
	Neither agree nor disagree	197	15
	Agree	575	44
	Strongly agree	450	34
It is important to have legislation that ensures animal care is adequate	Strongly disagree	21	1
	Disagree	25	1
	Neither agree nor disagree	126	9
	Agree	557	42
	Strongly agree	571	43
Animal protection organizations are important in ensuring animals are adequately cared for	Strongly disagree	19	1
	Disagree	31	2
	Neither agree nor disagree	119	9
	Agree	537	41
	Strongly agree	594	45

## Data Availability

The raw data has not been published or stored elsewhere but is available on request from F.C.

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
