# Peer review of "Chinese Public Attitudes towards, and Knowledge of, Animal Welfare"

_animals, 2021, doi:10.3390/ani11030855_

Round 1

Reviewer 1 Report

As authors mention, study of animal welfare in China is a very important and is much needed.  Since China is a vast country with much livestocks, understanding how they are treated is of interest to the world.  

In this study, they have surveyed over 1200 subjects with major focus on Inner Mongolia.  Inner Mongolia is very different from other area of China with a long history of nomadic tradition.  Even in urban areas, people are familiar with the nomadic tradition and their food resources depend on nomads.  Nomads take care of their animal very differently from other type of farms, thus their attitudes would be very different.  

In this respect, combining the Inner Mongolian data with other areas of China may not be appropriate. Unless the Inner Mongolian data is similar to the other data, these data should not be treated as data representing China. 

If there is a need to combine all the data, authors should provide solid reasons for doing so and representing it as data of China.

Treating the result of this study as Chinese data may be misleading. For example, comparing the Asian, or Chinese data reported by Phillips, et al. 2012 with the results of this data may not be appropriate.  If they do, they must show why it is possible to compare with Philipps' results.

 Mongolians have a long history of Nomads, living with animal.  The human animal relationship is quite different from other farming styles.  It would be of the interest to the readers by focusing just on Inner Mongolian data rather than combining them with other urban or rural data.

I recommend a major revision by focusing on Inner Mongolian  data which would be of much interest to many of the readers. The concept of animal welfare may be very different from Western farms.

Author Response

Responses to reviewer 1.

Reviewer 1 proposes an interesting idea. We would nevertheless like to keep this as an all-China presentation. Although the research was headquartered from Inner Mongolia we believe that we have sufficient responses from the rest of China to provide an overall valid perspective on the opinions of the whole of China. We have checked the data in the light of these comments and it was clear that the differences could not be attributed to province with any certainty. We have added a comment on this in the text. We have added a comment in the results section, in line 179 to clarify this:

Although most respondents were resident in the province of Inner Mongolia, there were no significant differences found for this subset of respondents for the responses received.

Additionally, the province of Inner Mongolia is not mono-cultural and includes among its population a large number of people whose origin is from elsewhere in China. 

The suggestion of the perspectives and opinions of nomadic peoples is however a very interesting one, and aligns with a previous pilot study by one of the co-authors on the opinions of nomadic peoples further west. A similar study looking specifically at the opinions of those residents of traditional nomadic societies in Inner Mongolia, Mongolia and Kazakhstan would be very interesting. If the reviewer would be interested in collaborating on such a study he/she would be very welcome to contact us.

Reviewer 2 Report

L61: there is one "are" to much in this sentence.

LL74/75: Meaning of the sentence is unclear

L96: I dont understand these findings. If the environment (housing condition) is favourable for an animal, then this meant the the animal welfare is improved, too. Please explain more clearly.

Table 3, Question 3 and Table 3 Question: I think the answer to these questions is very much dependent on whether the person actually owns or cares for an animal. People, who do not own or care for animals have no reason to learn about animal caring and it is probbably not important for them as a person. This should be considered in the interpretation of the results. Why did you not ask about animal ownership in the questionnaire?

Fig.2a/b: y-Axis Lable is missing

Table 4: Why didnt you give the participants the opportunity to give their own reason instead of asking for agree/disagree of set answers? You might have discovered reasond, you could not think of from a scientific perspective.

Table 4: If the aim of this question was to rank or compare your set of answers, why was this questions designed that way? Why, for example, did you not ask to rank your set of answers from 1-8?

Author Response

Responses to reviewer 2,

L61: there is one "are" to much in this sentence.

Yes, this is corrected, text changed as below.

Within the same report, it was further noted that animal welfare was more important for female respondents than for male respondents, and also for younger respondents [5]. Food-producing animals make up the majority of animals that are managed by humans globally [6], and animal farming systems are accused of inefficient use of scarce resources, in particular feed, water and land.

LL74/75: Meaning of the sentence is unclear

Yes, this is corrected. Text changed as below.

The Chinese livestock industries have experienced a variety of major animal epidemics, such as SARS, avian influenza, foot and mouth disease and more recently, African swine fever, all of which necessitated large numbers of animals being removed from the supply chain with considerable impact on both the livestock market and animal welfare.

L96: I dont understand these findings. If the environment (housing condition) is favourable for an animal, then this meant the the animal welfare is improved, too. Please explain more clearly.

Sentence rewritten for clarity. Text changed as below.

Platto et al [21] asked Chinese farmers to rate several different priorities for action on farms, for example, provision of better flooring to promote hoof health or better lying areas, and the improvement of animal welfare was rated third, the most important being the farmer’s own well-being.

Table 3, Question 3 and Table 3 Question: I think the answer to these questions is very much dependent on whether the person actually owns or cares for an animal. People, who do not own or care for animals have no reason to learn about animal caring and it is probbably not important for them as a person. This should be considered in the interpretation of the results. Why did you not ask about animal ownership in the questionnaire?

Well, that may be true, but actually the purpose of these questions was to ask respondents from where they were informed about animal welfare. Ownership of animals can include a range of disparate animal relationships (pet ownership, guard dog ownership, production animal ownership. For clarity we chose to simply ask about the origin of their information.

Fig.2a/b: y-Axis Lable is missing

Added as suggested

Table 4: Why didnt you give the participants the opportunity to give their own reason instead of asking for agree/disagree of set answers? You might have discovered reasond, you could not think of from a scientific perspective.

Yes, that is true, But we wanted, for statistical purposes, to keep the responses into discrete categories that we could then analyse.  It would also have added to the length of the questionnaire for each participant. Nevertheless, it is a good idea, and for future studies we will incorporate more opportunities for free text answers

Table 4: If the aim of this question was to rank or compare your set of answers, why was this questions designed that way? Why, for example, did you not ask to rank your set of answers from 1-8?

Well, ranking might have been an idea, but then for each category we would only have known their relative importance, whereas in this way we could get an absolute assessment of each of the categories, not relative to the other options.  We chose to present them in declining order of agreement for clarity of understanding and comparison for the reader.

Round 2

Reviewer 1 Report

I was concerned about the demographics of the subjects used in this study.  Since they have added the comment indicating that Inner Mongolian data did not differ from other subjects, I think the issue is explained and is no longer the problem.  They have also mentioned at the end for the limitation of the study.  

I find this kind of research is very important, particularly in rapidly changing country.  Changes in coming years in China may be a hint to other  developing countries in promoting animal welfare issues.